**Elemental Stoichiometry of Particulate Organic Matter across the Atlantic Ocean**

Adam J. Fagan[1], Tatsuro Tanioka[1], Alyse A. Larkin[1], Jenna A. Lee[1,] Nathan S. Garcia[1], & Adam C. Martiny[1,2,*]

[1]Department of Earth System Science, University of California, Irvine, CA, USA

[2]Department of Ecology and Evolutionary Biology, University of California, Irvine, CA, USA

*Corresponding Author: amartiny@uci.edu

**Abstract:**
Recent studies show that stoichiometric elemental ratios of marine ecosystems are not static at Redfield proportions but vary systematically between biomes. However, the wider Atlantic Ocean is under-sampled for particulate organic matter (POM) elemental composition, especially as it comes to phosphorus (i.e., POP). Thus, it is uncertain how environmental variation in this region translates into shifts in C:N:P. To address this, we analyzed hydrography, genomics, and POM concentrations from 877 stations on the meridional transects AMT28 and C13.5, spanning the Atlantic Ocean. We observed nutrient-replete, high-latitude ecosystem C:N:P to be significantly lower than the oligotrophic gyres. Latitudinal and zonal differences in elemental stoichiometry were linked to overall nutrient supply as well as N vs. P stress. C:P and N:P were generally higher in the P-stressed northern region compared to southern hemisphere regions. We also detected a zonal difference linked to a westward deepening nutricline and a shift from N to P stress. We also evaluated possible seasonal changes in C:N:P across the basin and predicted these to be limited. Overall, this study confirms latitudinal shifts in surface ocean POM ratios but reveals previously unrecognized hemisphere and zonal gradients. This work demonstrates the importance of understanding how regional shifts in hydrography and type of nutrient stress shape the coupling between Atlantic Ocean nutrient and carbon cycles.

**Plain language summary:**
Climate change is predicted to influence the biological pump by altering phytoplankton nutrient distribution. In our research, we conducted comprehensive measurements of particulate matter concentrations during two large oceanographic field studies. We observed systematic variations in organic matter concentrations and ratios across the Atlantic Ocean, both latitudinally and longitudinally. Through statistical modeling, we determined that these variations are associated with differences in the availability of essential nutrients for phytoplankton growth. Our findings highlight the adaptive resource utilization among surface ocean plankton, which in turn modulates the interplay between the ocean's nutrient and carbon cycles.

**Key points:**
- There was systematic regional variation in POM concentrations and ratios across the Atlantic Ocean.
- Latitudinal variability in C:N:P is linked to the nutrient supply rate and N vs. P stress.
- Westward deepening isopycnals and nutricline and a shift from N to P stress correspond to zonal variability in C:N:P

## 1. Introduction

The efficiency of the biological pump is anticipated to be affected by climate change through alteration in phytoplankton nutrient allocation and the C:N:P ratio (Galbraith and Martiny, 2015). Nevertheless, the influence of ocean warming on this efficiency is still uncertain, carrying potential repercussions for the ecosystems and global carbon cycle (Kwon et al., 2022). Over the past few decades, studies have observed variability in marine plankton elemental composition and ecosystem elemental composition (Weber and Deutsch, 2010; Martiny et al., 2013b, a). Specifically, regions with nutrient-rich conditions have lower C:N:P ratios (equatorial, coastal, and temperate regions), and nutrient-poor conditions (subtropical gyre regions) have higher ratios (Martiny et al., 2013b, a). However, data compilations include variations in both sampling and analytical methodologies (Martiny et al., 2014) as well as have limited spatial coverage. Therefore, large-scale sampling efforts like Bio-GO-SHIP are quantifying ecosystem particulate organic matter (POM) concentrations and their elemental ratios utilizing consistent methodologies on a global scale (Tanioka et al., 2022; Clayton et al., 2022).

Studies focused on POM stoichiometry across ocean basins have been primarily limited to Bio-GO-SHIP cruises within the Indian Ocean (Garcia et al., 2018) and the Pacific Ocean (Lee et al., 2021). Both studies have observed high POM concentrations at higher latitudes and low concentrations within the gyres, with intermediate levels toward the equator. The stoichiometry had higher values in the gyres and lower values at high latitudes (Garcia et al., 2018; Lee et al., 2021). There have been two basins-wide transects across the Atlantic Ocean that have been used in a global synthesis (Tanioka et al., 2022) but have not been used in a study focused solely on the Atlantic. Along with the strong relationship with latitude, there is also strong correlation with nutricline depth, used as a proxy for nutrient flux, in the global synthesis. Localized studies at the Bermuda Atlantic Time-series (BATS) site or short transects along the western North Atlantic Ocean show an N:P ratio between 40–50 and C:N near Redfield proportions (~6.6) (Michaels et al., 1994; Michaels and Knap, 1996; Steinberg et al., 2001; Babiker et al., 2004; Cavender-Bares et al., 2001). In contrast, POM dynamics and especially N:P and C:P ratios are less understood within the NE Atlantic and South Atlantic Oceans as a whole.

The Atlantic Ocean has a unique dynamic, being singularly/ co-limited by nitrogen and phosphorus respectively to the north of the equator and predominantly nitrogen-limited south of the equator (Cotner et al., 1997; Mather et al., 2008; Browning and Moore, 2023). In phosphorus co-limited regions, N:P and C:P are often elevated from frugal phosphorus use, supported by the well-sampled NW Atlantic Ocean (Galbraith and Martiny, 2015; Lomas et al., 2010, 2022). As a response to the nutrient limitation, phytoplankton can express specific genes that will allow for greater uptake of a nutrient. Gene expression and preferential uptake could influence cellular C:N:P within phytoplankton. Nitrogen limitation is more widespread in the South Atlantic Ocean, but no study has quantified ecosystem C:N:P here (Mather et al., 2008; Ustick et al., 2021). Temperature has been known to influence the concentration of cellular phosphorus in phytoplankton, with increasing in C:P with warmer temperatures, however C:N remains unchanged (Yvon-Durocher et al., 2015). The underlying mechanism for this relationship is not fully understood but hypothesized to be from either an increase in carbon uptake over phosphorus, an increase in nutrient use efficiency, or translation compensation theory (few P-rich ribosomes are required for protein synthesis) (Tanioka and Matsumoto, 2020). The availability of

nutrients generally follow inverse patterns of C:N:P, with increasing nutrients leading to a decrease in C:N and C:P and vice-versa (Galbraith and Martiny, 2015; Tanioka and Matsumoto, 2017). However, such environmental variation in the Atlantic Ocean elemental stoichiometry remains largely unknown. Therefore, the broad environmental gradients in the Atlantic Ocean could result in significant regional ecosystem C:N:P shifts.

Here, we quantified suspended particulate organic carbon, nitrogen, and phosphorus concentrations along two Bio-GO-SHIP meridional transects: AMT 28 and C13.5 (Fig. 1), covering large parts of the Atlantic Ocean. We addressed two questions: (1) What are meridional, hemispheric, and zonal differences in POM concentrations and stoichiometry? And (2) What is the relationship between environmental factors and C:N:P? We hypothesize that differences in total nutrient supply and temperature are primarily responsible for the latitudinal gradient in C:N:P. In contrast, the type of nutrient stress will be important for hemispheric and longitudinal C:N:P shifts.

## 2. Methods
### 2.1. Cruise Transects

AMT 28 started in Harwich, UK (49° 38´ N/5° 30´ W), and ended in Mare Harbour, Falkland Islands (48° 12´ S/52° 42´ W), departing the 25 September 2018, and ending the 27  October 2018. C13.5 started in Cape Town, South Africa (34° 22´ S/17° 18´ W), and ended in Norfolk, VA (36° 5´ N/74° 34´ W) (Fig. 1), departing the 21 March 2020, and ending the 16 April 2020. C13.5 was set to go 45° S  and collect samples along the eastern boundary of the South Atlantic Ocean. Due to COVID-19 quarantine restrictions, it was redirected to a port in Virginia. Fortuitously, this redirect allowed sample collection across the eastern South Atlantic Ocean and the western North Atlantic Ocean.

### 2.2. Sample collection

Seawater for the POM was collected from the underway flow–through system for both cruises at a depth of approximately 5 m. This method involved initially passing water through a 30 µm nylon mesh to remove the stochastic presence of large particles from the samples (Lee et al., 2021). We then collected 3 to 8 L of filtered water in 8.5 L plastic polycarbonate carboys (Thermo Fisher Scientific, Waltham, MA). The carboys were placed at a 45° angle to prevent particles from settling below the nozzle. Next, particulate organic carbon (POC)/ nitrogen (PON), and phosphorus (POP) samples were filtered onto 25 mm pre-combusted GF/F (500° C for 5 hours)(nominal pore size of 0.7 µm) (Whatman, Florham Park, NJ) (POC/PON are on the same filter). POP filters were rinsed with 5 ml of 0.17 M $Na_2SO_4$ to remove traces of dissolved phosphorous from the filter. Finally, we stored all filters in pre-combusted aluminum packets and placed them in a -80° C freezer during the cruise, a -20° C cooler for shipping, and back to a    -80° C freezer until analysis. Between sample collections, the carboys and tubing were rinsed with 30 µm filtered sample water just prior to collection.

We collected single samples of POC/PON and POP hourly for AMT 28. For the C13.5 transect, POC/PON and POP samples were collected in triplicate every 4 to 6 hours. Water collection for C13.5 was done at the peak and trough of the diel cycle, ~06:00 and ~20:00 LT, respectively, and with one to two collections in between those times.

### 2.3. Particulate organic matter determination

*2.3.1. Particulate organic phosphorus (POP) assay*

POP was analyzed using a modified ash-hydrolysis protocol (Lomas et al., 2010). Filters were placed into acid-washed/pre-combusted glass vials with 2 ml of 0.017 M $MgSO_4$ and covered with pre-combusted aluminum foil. The vials were placed in an incubator for 24 hours at 80 to 90° C and then combusted for 2 hours at 500° C. After cooling, 5 ml of 0.2 M HCl was added and incubated at 80 to 90° C for 30 minutes. The supernatant was collected, and the vials were rinsed with 5 ml of Milli-Q water. The rinse water was collected and added to the supernatant. 1 ml of mixed reagent (2:5:1:2 parts ammonium molybdate tetrahydrate (24.3 mM), sulfuric acid (5 N), potassium antimonyl tartrate (4.1 mM), and ascorbic acid (0.3 M) were added to the supernatant and left in the dark for 30 minutes. Samples were analyzed on a spectrophotometer at a wavelength of 885 nm using a potassium monobasic phosphate standard (1.0 mM-P). The detection limit for POP measurements was ~0.3 µg.

*2.3.2. Particulate organic carbon/nitrogen (POC/PON) assay*

POC/PON are measured using the same filter. The POC/PON samples were processed in the lab at UCI using a JGOFS protocol (Ducklow and Dickson, 1994). POC/PON samples were dried in an incubator at 55° C for 24 hours. They were then moved to a desiccator with concentrated HCl fumes for 24 hours to remove inorganic carbon. The samples were then re-dried at 55° C for 24 hours before being packaged into pre-combusted tin capsules (CE Elantech, Lakewood, NJ). The packaged filters were analyzed on a CN FlashEA 1112 Elemental Analyzer (Thermo Scientific, Waltham, MA) with atropine and acetanilide standards. POC and PON measurements had a detection limit of ~2.4 µg and ~3.0 µg. Settings for the FlashEA had an oxidative reactor temperature of 900° C, a reduction reactor temperature of 680° C, and an oven temperature of 50° C. Oxygen introduced to the oxidative reactor lasted seven seconds allowing temperatures to reach 1800° C for sample combustion. A leak test needed to fall below 5 ml min$^{-1}$ before samples were analyzed to minimize sample loss.

## 2.4. Nutrient availability, biogeography, and biological properties
*2.4.1. Nutricline depth*

The nutricline depth was determined as the 1 µM nitrate depth horizon (Garcia et al., 2018; Cermeño et al., 2008). Nutricline depth was regarded as a proxy for nutrient supply to the surface, with a shallow nutricline representing a high flux of nutrients and vice versa for a deep nutricline. The nutricline depth with respect to the 1/16 µM phosphate depth horizon was also investigated but found to be nearly identical to that of nitrate. For AMT28, nitrate concentrations were quantified as previously described from CTD casts along the transect (Swift, 2019). Nitrate concentrations were then interpolated using DIVA implemented in Ocean Data View (v5.5.2) (Schlitzer, 2019). For C13.5, we used the seasonal average nitrate depth profiles from 2018 of the World Ocean Atlas at one-degree spatial resolution to determine nutricline depths. This approach was necessary as the logistical issues related to COVID-19 quarantine restrictions prevented us from collecting onboard CTD measurements. Linear interpolation for each profile within the one degree was performed to estimate the nutricline depth.

*2.4.2. Delineation of Regions*

The regions under consideration for this study are the Eastern Temperate North Atlantic (ETNA)
[Lat. 49.6˚N-43.2˚N] Western North Atlantic Gyre (WNAG) [Lat. 34.5˚N-19.8˚N], Eastern
North Atlantic Gyre (ENAG) [Lat. 43.0˚N-18.1˚N], Western Equatorial (WEQ) [Lat. 17.9˚N-
5.9˚S], Eastern Equatorial (EEQ) [Lat. 17.8˚N-5.9˚S], Western South Atlantic Gyre (WSAG)
[Lat. 6.0˚S-34.0˚S], Eastern South Atlantic Gyre (ESAG) [Lat. 6.2˚S-33.0˚S], Western
Temperate South Atlantic (WTSA) [Lat. 34.1˚S-48.2˚S], and Eastern Temperate South Atlantic
(ETSA) [Lat. 33.9˚S-41.5˚S] (Fig. 1). These boundaries are determined using inflection points
along the nutricline depth and the temperature profile.
*2.4.3. Cell size*
Cell size was determined by the conversion of cell count, collected during CTD casts (AMT28)
at the top 200 m of the water column. Flow cytometry samples (63 stations, 755 samples) were
co-collected with the POM samples used in this study. Cell count was determined using two
methodologies. The first method was collected without a filter and utilized an inverted
microscope to estimate cell abundance and phytoplankton species composition (Utermöhl, 1958).
This allowed for the estimates of diatoms, dinoflagellates, and coccolithophores. The second
method measured cells using a Becton Dickinson FACSort flow cytometer to measure
*Prochlorococcus, Synechococcus,* and pico-eukaryotes. Combining these two methods of
collection allowed for a complete survey of phytoplankton groups.
Conversion of cell count to biomass (fg C cell) was done following the methodology
from Moreno et al., 2022. Photoautotrophs were categorized into *Prochlorococcus,*
*Synechococcus,* pico-eukaryotes, nano-eukaryotes, coccolithophore, and cryptophytes. Each cell
type had a specific conversion factor in determining its biomass. Using a Monte Carlo approach,
95% confidence interval around cell size was determined using a normal distribution based on
the mean and standard deviation. Then, a randomly chosen conversion factor was applied to each
type. Allowing for 1000 runs, we estimate a 95% confidence interval (Moreno et al., 2022).
*2.4.4. Metagenomics-informed nutrient stress*
Metagenomically informed nutrient stress utilizes a subset of data from Ustick et al., 2021,
utilizing the genome content of *Prochlorococcus* from the Atlantic Ocean. These metagenomic
samples (276) were co-collected with the POM samples, across both transects, used in this study.
Based on variation in *Prochlorococcus* population gene content, this study identified genes
associated with nitrogen and phosphorus nutrient stress types. The severity of nutrient stress was
quantified by calculating the frequency of nutrient acquisition genes within *Prochlorococcus*
single-copy core genes and attributes the frequency to the genetic adaptation for overcoming
nutrient stress type and severity. Moving forward the use of nitrogen/phosphorus gene index will
refer to this calculation of nutrient stress. Although based on *Prochlorococcus*, there is a
significant overlap between this genetic index of nutrient stress and both Earth System Models
and whole community nutrient addition assays (Ustick et al., 2021).
Ustick et al., 2021 associated *Prochlorococcus* gene occurrences with different
environmental nutrient stress conditions. They separated the genes by nutrient type (nitrogen,
phosphorus, and iron) and nutrient stress severity (low, medium, and high). Our study utilizes the
high-stress severity for nitrogen and phosphorus. Iron has a more indirect influence on the

C:N:P, than nitrogen and phosphorus, which is why it will be omitted from this study. The stress severity associated with medium or low stress either followed the same pattern as the high nutrient stress or had no pattern at all, respectively, which is why this is also omitted. The function of the genes associated with high gene index are *focA, moaA-E, moeA, napA, narB, nirA (*for nitrogen) and *phoA, phoX* (for phosphorus). The functions of these genes are for the assimilation and uptake of nitrite and nitrate, and production of alkaline phosphatase.

*2.4.5 N\* Derivation*

The derivation from Redfield nutrient concentration (N\*) at a depth of 200 m was calculation:

$$N^*_{200} = [NO_3]^{-1}_{200} - 16[PO_4]^{-3}_{200}$$

A negative/ declining value would be indicative of nitrogen stress, while a positive/ increasing value would indicate phosphorus stress.

**2.5. Data analysis**

Data analysis was conducted using Matlab R2021b (MathWorks). An ANOVA analysis with a posthoc Tukey test was used to determine the relationship between the selected regions for environmental conditions and POM. The C:N:P ratios underwent a log transformation to achieve a normal distribution before the ANOVA analysis (Isles, 2020). Using R ver. 4.1.2 (R Core Team, 2021), we used generalized additive models (GAM) with package *mgcv* (v1.8) (Wood, 2017) to explain the strength of four variables in determining C:N:P (temperature, nutricline depth, nitrogen gene index, and phosphorus gene index).

## 3. Results

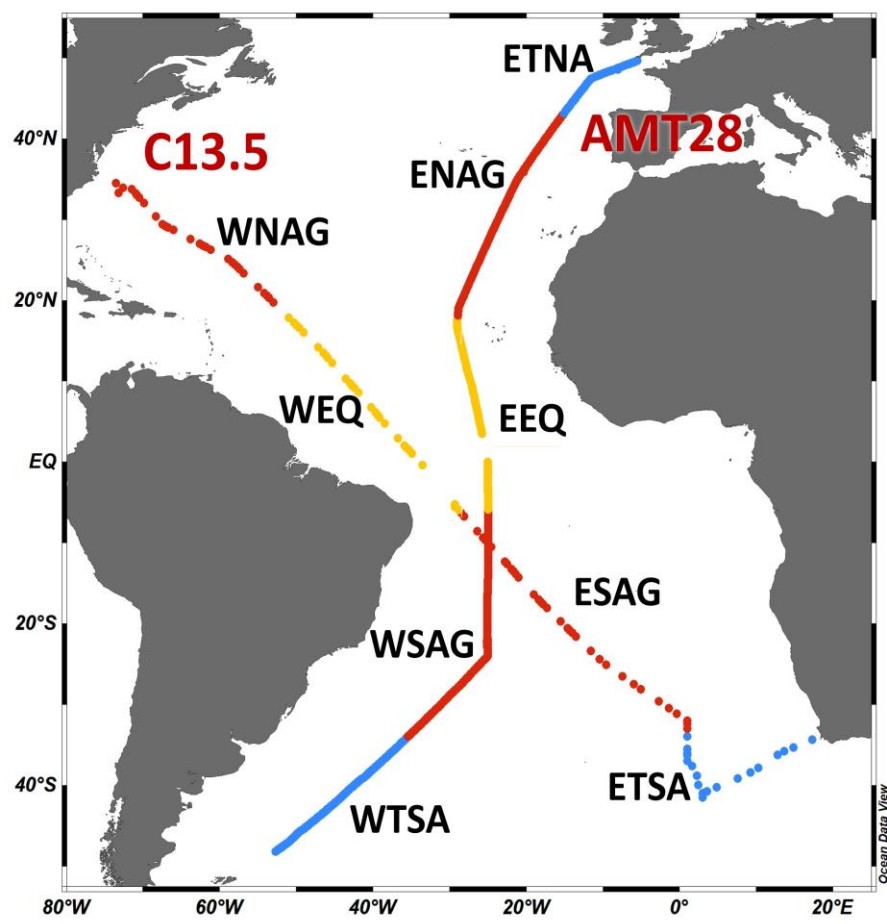

**Figure 1.** Map of oceanographic cruise transects AMT 28 (September 25 to October 27, 2018, n = 765) and C13.5 (March 21 to April 16, 2020, n = 112). Different oceanographic regions are separated using nutricline and temperature profiles (WTSA = Western Temperate South Atlantic, ETSA = Eastern Temperate South Atlantic, WSAG = Western South Atlantic Gyre, ESAG = Eastern South Atlantic Gyre, WEQ = Western Equatorial, EEQ = Eastern Equatorial, WNAG = Western North Atlantic Gyre, ENAG = Eastern North Atlantic Gyre, ETNA = Eastern Temperate North Atlantic). Colors delineate temperate (blue), subtropical (red), and equatorial upwelling regions (yellow).

POM concentrations, temperature, and nutricline profiles exhibited unique characteristics to each oceanographic region. Between the two transects, POC, PON, and POP concentrations were strongly correlated (r = 0.68, 0.71, and 0.70, respectively; $p < 0.001$) (Fig. 2a and S1). All POM pools had peak concentrations at high latitudes, troughs in the subtropical gyres, and intermediate concentrations at the equator. In high latitude temperate regions (WTSA, ETSA, and ETNA), POC (and overall POM) was significantly elevated (4.6 to 5.3 µM; $p < 0.05$) compared to all other regions (Equatorial: 2.8 µM, Gyre: 1.6 to 2.1 µM) (Fig. 2a, Fig. S2). POM concentrations also showed a zonal difference. There were higher concentrations of POM in the western regions compared to the eastern region of the Temperate South Atlantic, whereas the opposite was seen in the subtropical gyres (Fig. 2a and Fig. S2). At ~10° S, C13.5 and AMT 28 cross paths, we used a 1° cell centered on the intersection (using 9 samples), to find the difference between the POC, PON, and POP of the two cruises was 0.2%, 5.7%, and 10.6% respectively, indicating that seasonal variability between the had the greatest impact on POP. However, one sample is the cause of most of the error, within PON and POP, removing the

sample the difference becomes 2.9%, and 2.1%, respectively. Temperature peaked equatorially (~ 28° C) for both transects and declined with increasing latitudes (Fig. 2b). We observed minor variation in the meridional temperature profile linked to the difference in the seasonal timing for each cruise, leading to a slight southward shift in peak temperature during C13.5. Nutricline profiles for both transects were similar, with the deepest nutricline in the gyres and shallowest at high latitudes and the equator (Fig. 2c). Zonal variability in the nutricline depth was apparent, with the deepest values in the western side (135 to 150 m) compared to the eastern side of the gyres (114 to 116 m) (Fig. S2). Thus, we observed a robust meridional gradient in POM concentrations and environmental conditions but also a zonal gradient in nutricline depth in the oligotrophic subtropical gyres.

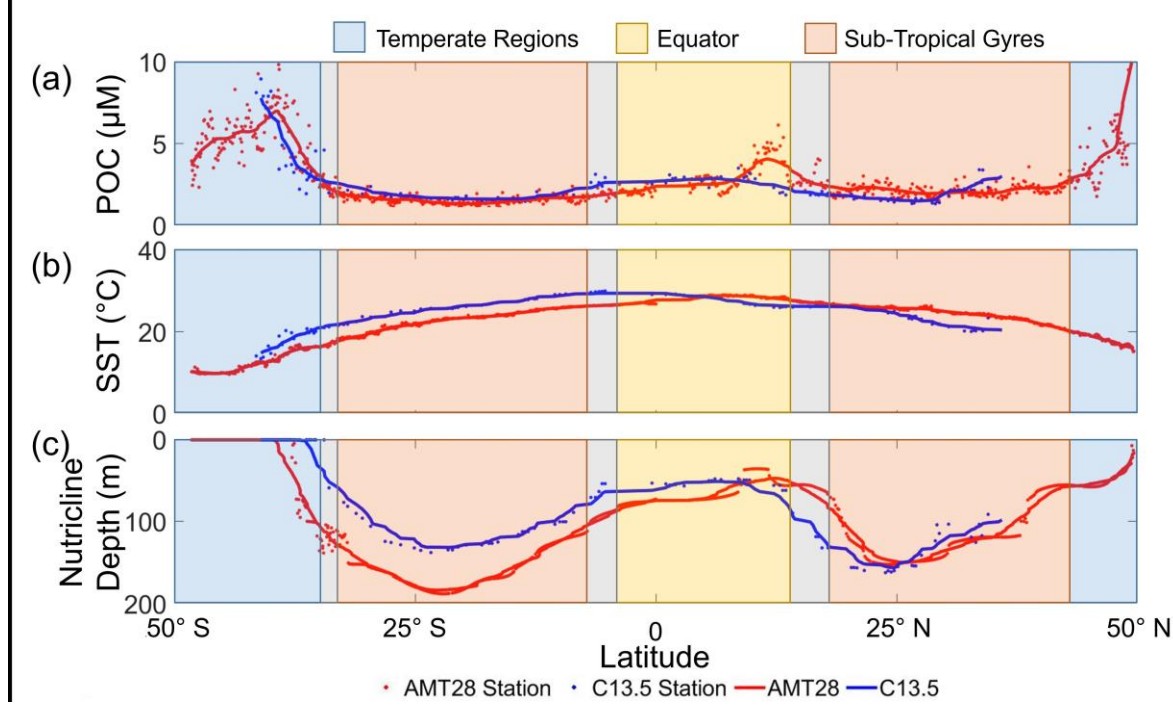

**Figure 2.** Meridional variability in POC concentrations and environmental conditions for AMT28 (boreal fall) and C13.5 (boreal spring). (a) Averaged surface POC concentrations, (b) surface temperature, and (c) nutricline depth presented as $Z_{nitrate} > 1$ μM. The trend lines represent the moving average of samples for AMT28 (red/ n=50) and C13.5 (blue/ n=20) transects. Background colors indicate broad oceanographic regions separated by latitude (blue = Temperate, red = Subtropical, yellow = Equatorial upwelling regions). Grey spaces between regions represent the difference in boundaries between the two transects.

We observed distinct latitudinal, zonal, and hemispheric C:N:P variability (Fig. 3). First, we detected peak ratios in the subtropical gyres, troughs in the high latitudes, and intermediate values at the equator for C:N, C:P, and N:P, matching patterns seen globally (Martiny et al., 2013b). In the subtropical gyres, averaged C:N values were noticeably elevated (7.0 to 7.6) compared to the other regions (Temperate: 6.0 to 7.2, EQ: 6.6 to 6.8) (Fig. 3a). C:P followed the same trend as C:N, with subtropical gyre regions being higher (148 to 208) than the other regions (Temperate: 122 to 158, EQ: 136 to 161) (Fig. 3b). N:P showed parallel changes to C:P except the South Atlantic Gyre showed a N:P range encompassing those of all other regions (20.1 to 29.2) (Fig. 3c). Second, azonal gradient was detected, whereby C:N was higher in the eastern side of the South Atlantic Ocean compared to the western side (Fig. 3D). However, this zonal

gradient was not observed in other regions. C:P also showed an opposite zonal trend with higher
values on the western side, albeit only significantly different in the northern hemisphere (Fig.
3e). N:P showed the highest zonal variation. This ratio was significantly higher on the western
(21.4) compared to the eastern side (17.1) of the South Atlantic Subtropical Gyre (Fig. 3f),
converging at ~10° S and again elevated on the western side (29.2) compared to the eastern side
(24.8) of the North Atlantic Subtropical Gyre (Fig. 3f). Again, using the 1° cell centered on this
intersection, we determined C:N, C:P, and N:P had a 5.8%, 12.1%, and 5.9% difference,
respectively, between the two cruises. One sample is the cause of a majority of the error, with its
removal, the difference becomes 2.6% for C:N and 1% for the rest. Third, there was also a
hemisphere bias, whereby C:P, and N:P were elevated in the northern hemisphere and C:N
somewhat higher in the southern hemisphere. In summary, we saw clear latitudinal, zonal, and
hemisphere gradients in C:N:P across the Atlantic Ocean.

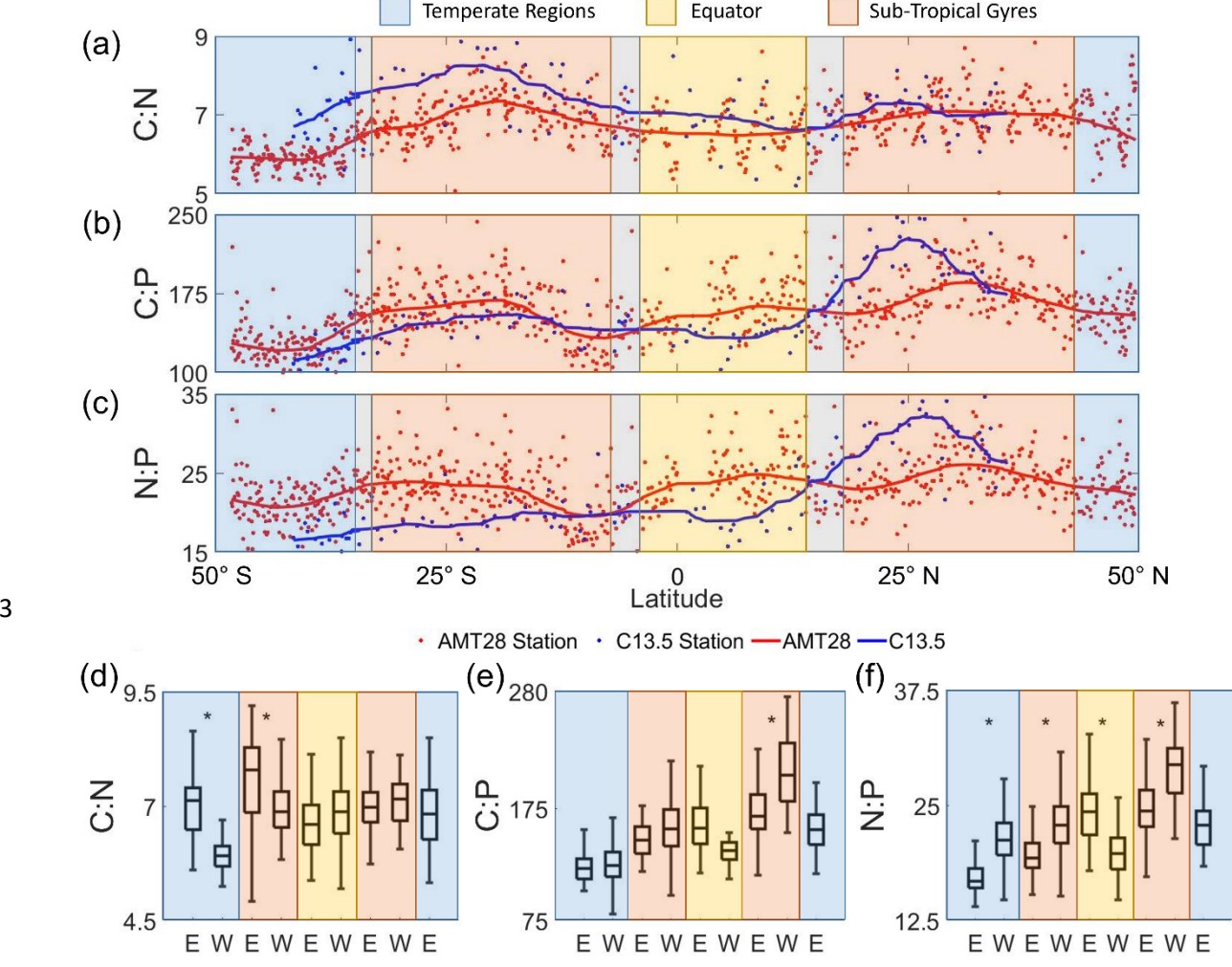


**Figure 3.** Latitudinal and regional shifts in POM stoichiometry. (a-c) Averaged observed surface C:N, C:P, and N:P.
The trend lines represent the moving average of samples for AMT28 (red) and C13.5 (blue) transects. Linear
regression line representative of all samples along the transects (black). (d-f) Regional C:N, C:P, and N:P
represented by boxplots, where data were separated by latitude and longitude (E = East. W = West). Significant
zonal (east-west) differences are denoted with * above plot based on Tukey posthoc significant difference test ($p$ =

 0.05). For all boxplots, a central black bar of the box represents the median value. The whiskers signify the range
(min, max) of values excluding outliers.
The variability of C:N:P across regions can be partially explained when investigating N*
at 200 m for AMT 28. Across the transect, N* has a positive value from 10° N to 50° N, with the
remaining regions having a negative value (Fig. S3). As N* decreases from north to south, the
environment becomes more nitrogen-stressed. When comparing N* and N:P directly, there is
only a weak correlation (r = 0.48, $p < 0.001$). Beyond the general increasing value of both N*
and N:P from the south to the north, the features of the two plots do not line up directly. Rather it
would appear that the peaks in N* more closely align with the troughs in N:P and vice versa.
Using flow cytometry cell counts, we were able to determine the concentration and total
biomass of separate species of photoautotrophs at each station for AMT 28. From this,
*Prochlorococcus* was determined to make up > 93% of the community in the subtropical gyres
and equator, and over 50% of the total biomass. 67% of the northern temperate region
community consisted of *Prochlorococcus* but only 10% of the biomass*,* and the South Temperate
Atlantic Ocean was the only region without *Prochlorococcus* being the most abundant at 12% of
the community and 1% of the biomass (Fig. S4). With the fractional biomass of the six
phytoplankton size groups, we used a linear regression model to link to C:P along the transect.
The regression model was able to describe the general characteristics of the in situ samples but
failed to capture the detailed transitions ($R^2 = 0.23$ $p < 0.05$) (Fig. S5). While only being able to
capture the general characteristics of the in situ samples and the dominant biomass of
*Prochlorococcus* across the Atlantic, we found that the use of gene-specific nutrient stress of
*Prochloroccus* to be an acceptable driver of the variability of C:N:P within GAM.
The influence of phytoplankton composition, temperature, nutricline depth, and
metagenomically assessed nitrogen and phosphorus stress (gene index) were tested as drivers of
stoichiometry using a general additive model (GAM) (Fig. 4). Using GAM, we determined
temperature and the nutrient gene indices captured 67% and 56% of the total deviance for C:P
and N:P, respectively. For C:P, nutricline depth and phosphorus gene index accounted for 52.5%
of the total (31.3% and 21.2%, respectively). For N:P, nutricline depth and phosphorus gene
index accounted for 45% of the total (24.6% and 20.7%, respectively). We could only explain
30% of the total deviance for C:N, with the temperature being the most significant contributors
(13% and 11%, $p < 0.001$ and $p < 0.01$ respectively). For C:N:P, nutricline depth was the
dominant contributor to the latitudinal variability for two of the three ratios, being the second
most dominant in the third, when investigating the entire basin (Fig. 4). When dividing the
Atlantic Ocean into eastern and western boundaries, the four drivers tested were able to explain
the variability of C:P and N:P more accurately in the western side (81% and 63% respectively)
and C:P in the eastern side (38%) (Fig. S7 and S8, Table S2). From this division the dominant
drivers remained nutricline depth and temperature for C:P and N:P, and became the dominant
driver of C:N. While the drivers for C:N individually have a maximum of 7% difference between
each other on either side of the Atlantic Ocean, the regional focus is able to interpret changes in
drivers that an ocean-wide analysis would determine to be different.

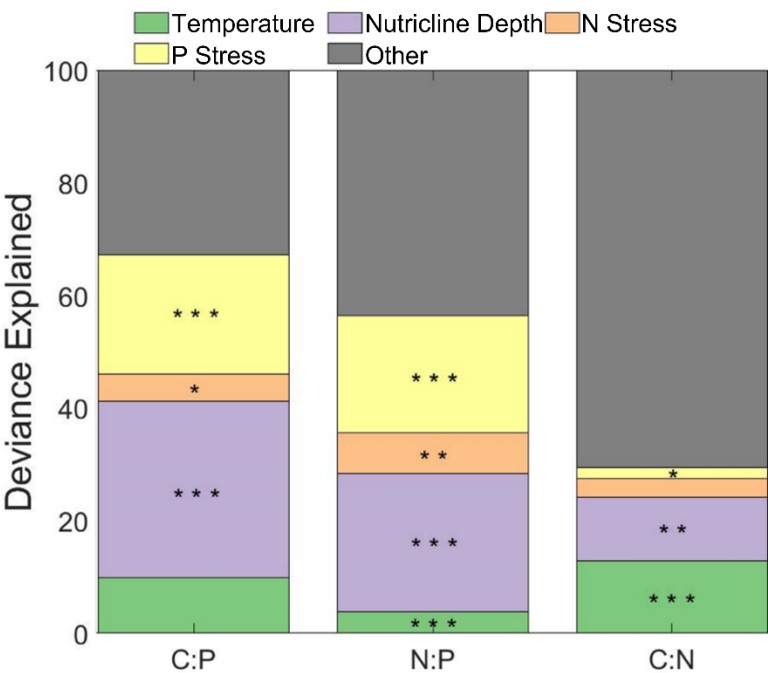

**Figure 4.** Influence of environmental factors on stoichiometry. Stars indicate the significance of smooth terms used for Generalized Additive Models (GAM). *** = $p < 0.001$, ** = $p < 0.01$, * = $p < 0.05$. Green represents the influence of temperature, purple represents the influence of nutricline depth, orange represents the nitrogen stress, yellow represents the phosphorus stress, and grey represents the remaining factors of influence on the variability of C:N:P. N and P stress are reflective of the nutrient gene index, which is quantified by calculating the frequency of the nutrient acquisition genes within *Prochlorococcus* single-copy core genes. The frequency is attributed to the genetic adaptation for overcoming nutrient stress type and severity.

A zonal gradient in nutricline depth and metagenomically assessed nitrogen and phosphorus stress matched C:N:P shifts (Fig. 3d–f). Nutricline depth was significantly deeper (*p* < 0.05) in the western part of subtropical gyres in both hemispheres (Fig. S2). Furthermore, there was a westward shift from nitrogen towards phosphorus stress (Fig. S6). This zonal shift in nutrient availability corresponds to a similar increase in C:P from 174 to 207 and N:P from 24.8 to 29.2 towards the western side of the oligotrophic gyres (Fig. 3e, f). In parallel, C:N showed the opposite trend declining from 7.6 on the eastern to 7.0 on the western side, matching a shift from nitrogen to phosphorus stress (Fig. 3D). GAM analyses conducted separately for western and eastern basins corroborated these observations, highlighting that the relative importance of shifting nutrient stress (Fig. S7–9). In summary, zonal variability in nutrient stress, described by a westward deepening nutricline and increased phosphorus gene index, may regulate a zonal change in C:N:P.

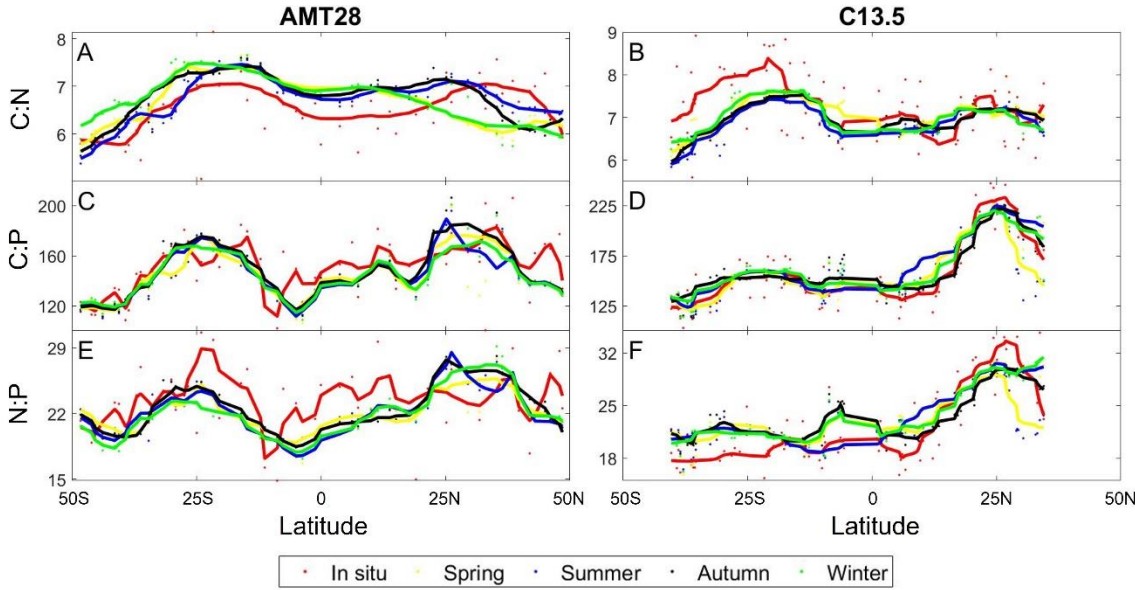

**Figure 5.** Predicted seasonal variability of stoichiometry across the Atlantic Ocean. Observed compared to predicted
seasonal C:N for AMT28 (A) and C13.5 (B). Observed compared to predicted seasonal C:P for AMT28 (C) and
C13.5 (D). Observed compared to predicted seasonal N:P for AMT28 (E) and C13.5 (F). Dots are discrete samples
and the lines are moving averages over ten samples. AMT28 occurred during the fall 2018 and C13.5 during the
spring 2020. In situ samples are red, predicted Spring is yellow, predicted Summer is blue, predicted Autumn is
black, and predicted Winter is green.

We assessed the potential impact of seasonal environmental changes for C:N:P across the
Atlantic Ocean. Seasonal environmental changes were characterized as shifts in nutricline depth
and temperature, while assuming a stable biogeography of nitrogen vs. phosphorus stress (Fig.
5). This assumption is the result of only having gene stress information from the season samples
were collected in. As a control, we saw a significant correlation between the observed and
predicted C:N:P for the season matching the cruise occurrence (Table S3). However, the
statistical model did not predict high C:N in the eastern South Atlantic Ocean and overestimated
N:P in the equatorial and western South Atlantic Ocean. C:N:P ratios were predicted to be
mostly stable across seasons. Although we detected shifts in C:N near the north sub-tropical
convergence zone (~18° C) reflecting an expansion and contraction of oligotrophic conditions
(Fig. 5a). The introduction of more dynamic biogeography of nutrient stress will be necessary to
predict a more accurate seasonal variability of C:N:P across the Atlantic Ocean. However, from
data available our statistical model predicted a mostly stable, seasonal C:N:P across the Atlantic
Ocean.

**4. Discussion**

There was clear latitudinal variability in POM concentrations and stoichiometry across the
Atlantic Ocean. We detected a high POM concentration and low C:N:P at higher latitudes, low
POM concentrations and high ratios in the subtropical gyres, and intermediate values near the
equator. This meridional gradient in POM concentrations and ratios corresponded to parallel
changes in nutricline depth and thus likely linked to the overall nutrient supply. Similar gradients
in concentrations and ratios have been detected in the Indian Ocean (Garcia et al., 2018), the
Pacific Ocean (Lee et al., 2021), and in a global synthesis (Martiny et al., 2013b). Thus, our
observations add further support to systematic biome shifts in C:N:P across major ocean basins.
Despite having similar gradients, the North Atlantic Ocean appears to be relatively unique with
higher C:P and N:P ratios in the northern hemisphere compared to the south. Both the North
Atlantic and the Indian Ocean's Bay of Bengal have comparable aeolian iron inputs, however,
North Atlantic Ocean has an increase in $N_2$–fixation, which increases the N:P nutrient supply
ratio, leading to widespread phosphorus stress (Capone, 2014; Schlosser et al., 2014; Ussher et
al., 2013). The Bay of Bengal does not have significant $N_2$–fixation nor a significant change in
C:P or N:P ratios (Garcia et al., 2018; Löscher et al., 2020). This lack of $N_2$–fixation is possibly
the result of stress from another micronutrient for $N_2$–fixers.
Focusing on the influence of P stress, there is an increase in phytoplankton elemental C:P
and, to a lesser extent, N:P throughout much of the North Atlantic Ocean. POP has a minimum
concentration in the western North Atlantic Ocean (Fig. S1), suggesting that the parallel changes
in N:P and C:P are caused by lower POP concentrations. Iron inputs decrease across the North
Atlantic Ocean from east to west, with a majority of the POP concentrations following the same
trend (Mahowald et al., 2005). While there is an increase in POP concentrations for C13.5, part
of this is attributed to coastal upwelling. Had C13.5 continued North it is possible that the POP
concentrations observed in the lower half of the North Atlantic gyre would have continued.
Proposed explanations of this zonal difference result from a combination of vertical iron supply
and lateral circulation across the Atlantic Ocean (Martiny et al., 2019). In the South Atlantic
Ocean, aeolian iron inputs are significantly lower, as most dust is washed out at the Intertropical
Convergence Zone (Capone, 2014). $N_2$–fixation is hence suppressed (Wang et al., 2019),
allowing most of the southern hemisphere to display elevated N stress. This rise in nitrogen
stress likely causes the depressed PON concentrations (Fig. S1) and elevated C:N but depressed
N:P in much of the South Atlantic Ocean. Thus, the hemisphere deviation in C:N:P is
hypothesized to be driven by a causal link between iron inputs, $N_2$–fixation, and shifts between
the nitrogen and phosphorus gene index (Martiny et al., 2019).
An additional zonal gradient in C:N:P may be linked to the westward deepening of the
nutricline and a parallel shift from primarily nitrogen stress towards an increase in phosphorus
stress. Phosphorus stress is detected throughout the central North Atlantic Ocean based on both
the gene index and N* (Ustick et al., 2021), however both C:P and N:P are significantly higher
on the western side. Using the nutricline depth as a proxy of nutrient supply, the nutrient supply
appeared greater on the eastern side, in addition, aeolian nutrient inputs could relieve nutrient
stress towards the east, suppressing C:P and N:P ratios (Kremling and Streu, 1993; Mills et al.,
2004; Garcia et al., 2018; Neuer et al., 2004). The South Atlantic Ocean also has the east–west
variability for C:N:P, with C:N having the largest gradient. From the nutrient gene index and N*,
the South Atlantic Ocean is predominantly nitrogen stressed. Zonal shifts in C:N:P can be
explained by shallower nutricline depth and a higher nitrogen gene index in the eastern part and a
higher phosphorus gene index in the western part of the South Atlantic Ocean (Ustick et al.,
2021; Martiny et al., 2019). Thus, we observe zonal variability in POM concentrations and their
stoichiometric ratios, superimposed on the larger meridional and hemisphere gradients.
Nitrogen and phosphorus stress are assessed based on genomic changes and adaptation in
*Prochlorococcus* populations (Ustick et al., 2021). With *Prochlorococcus* being the most
abundant phytoplankton and that it forms most of the phytoplankton biomass in the gyres and
equatorial regions, and the northern temperate population, it is likely closely linked to the bulk
phytoplankton community physiological status (Fig. S4) (Marañón et al., 2000; Zwirglmaier et
al., 2007). Additionally, *Prochlorococcus* and *Synechococcus* express nearly identical responses
across a transect with regions of different nutrient stress (i.e., when *Prochlorococcus* had a high
phosphorus gene index, *Synechococcus* had a high phosphorus gene index as well) (Garcia et al.,
2020). Within the South Atlantic Ocean, the use of bioassays and deficiency calculations agree
with *Prochlorococcus* gene stress, being primarily nitrogen stressed, yet disagree within the
North Atlantic Ocean (Browning and Moore, 2023). While previous bottle experiments of
nutrient stress in the North Atlantic Ocean describe it as being dominantly or co-stressed by
nitrogen and phosphorus, respectively, the gene index describes the North Atlantic as dominantly
phosphorus stressed. This suggests that there is a significant difference between the different
assays in determining the nutrient stresses phytoplankton experience. This study focused on
factors that had a direct influence on C:N:P, we then chose to forgo using co-stressors of
nutrients or the use of iron stress. Along with direct influence, these samples match one-to-one
with the POM samples collected on the cruises.
It was determined through the use of GAM, that nutricline depth, phosphorus stress, and
temperature were the main drivers in the variability of C:N:P. These findings are similar to those
of a global synthesis that determined nutricline and gene index were the dominant drivers of
C:N:P variability within the tropical and subtropical regions (Tanioka et al., 2022). While their
pole–wards assessment determined that temperature was the dominant driver, the samples used
in this study fall primarily within tropical/ subtropical bounds (49 of 877 samples are outside of
this range). C:P and N:P generally agreed with this global model assessment, but C:N
temperature had a smaller influence globally than for the Atlantic Ocean. With the relatively
small amount of variance determined for C:N, it is possible that the northernmost samples had a
major impact on the determination of temperatures influence, as seen by Tanioka et al. (2022), in
which temperature was determined to be the most significant driver for the variance of C:N.
With respect to the other section of the GAM analysis, the factors with a more indirect
relationship to C:N:P could have a significant role, especially with C:N (i.e., the influence of iron
stress or light availability).
The predicted restricted changes in seasonal values of C:N:P were able to fall in the
middle to lower range of the observed seasonal averages of those observed at BATS,
representing the fall and winter seasons better than spring and summer (Singh et al., 2015). It is
worth noting that while the values were able to capture the lower range, the ratios measured
during C13.5 closest to BATS, were lower than the measured monthly averages. Since C13.5
was unable to take CTD measurements, the nutricline depth from WOA might not accurately
represent the actual nutricline depth during the transect, leading to potential changes in the
predictive seasonal values. The intersection point of the two transects (~10˚ S) also indicates
minimal seasonal influence as the POM and stoichiometric values despite collection occurring in
opposite seasons. Using the values predicted by GAM for the same parameters, there was less
than a 2% difference in C:N:P between fall and spring indicating that some of the assumptions
made with the predictors weakened the sensitivity of the model. Without this sensitivity, the
predictive model suggests that the observed biogeography of C:N:P is stable in most of the
central Atlantic Ocean,. In summary, we detect clear meridional, hemisphere, and zonal
gradients in elemental stoichiometry that correspond to changes in nutrient supply and stress
type, but additional factors may also provide a significant influence on regional shifts in C:N:P
across the Atlantic Ocean.
Our observations from the Atlantic Ocean have implications for predicting future changes
to the ocean carbon cycle. Recent models have suggested that C:N:P variability can 'buffer' the
effects of stratification and reduced nutrient supply on primary productivity and carbon
sequestration (Kwon et al., 2022; Tanioka and Matsumoto, 2017). Such models of C:N:P
variability have so far been tied to surface phosphate concentrations (Galbraith and Martiny,
2015). However, our observations from the Atlantic Ocean indicate that subtle shifts between
nitrogen and phosphorus stress can have additional impacts on the elemental stoichiometry. $N_2$–
fixation in the North Atlantic Ocean is likely responsible for part of the shift in nutrient stress
type. The hemispheric variability of nutrient stress suggests an additional role of iron supply in
regulating C:N:P. Thus, climate change may alter future patterns of C:N:P as the perturbation of
air–sea dynamics can modulate the strengths of boundary currents, the slope of a westward
nutricline (Kelly et al., 2010), or the aeolian deposition of iron (Krishnamurthy et al., 2010).
Such shifts in C:N:P could, in turn, have large impacts on global nitrogen fixation, primary
production, or carbon sequestration.

**Conflict of interest**
The authors declare no conflicts of interest relevant to this study.

**Acknowledgments**
We thank the Global Oceans Ship-Based Hydrographic Investigations Program (GO-SHIP) and
the Atlantic Meridional Transect Programme for facilitating this project. We extend a special
thanks to Andrew Rees, Glen Tarren, and the crew of the *RSS James Clark* and Leticia Barbero
and the crew of the *R/V Roger Revelle*. This research was funded by the National Science
Foundation (OCE-1848576 and 1948842 to ACM), NASA (80NSSC21K1654 to ACM), NOAA
(101813 Z7554214 to ACM), and Simons Postdoctoral Fellowship in Marine Microbial Ecology
(724483 to TT). The PML AMT is funded by the UK Natural Environment Research Council
through its National Capability Long-term Single Centre Science Program, Climate Linked
Atlantic Sector Science (grant number NE/R015953/1). This study contributes to the
international IMBeR project and is AMT contribution number XXX (number pending).

**Data availability statement**
The AMT data set presented here is publicly hosted by the British Oceanographic Data Centre
(https://doi.org/10.5285/b5900384-89f0-3a38-e053-6c86abc0409d). Hydrographic data from the
AMT28 transect are available (https://cchdo.ucsd.edu/cruise/74JC20180923). The particulate
organic matter data from the C13.5 transect are available here
(https://www.bco-dmo.org/dataset/868908). Hydrographic data from C13.5 data are available
(https://cchdo.ucsd.edu/cruise/33RO20200321). Nutricline depth for C13.5 is calculated from
gridded annual mean nitrate data from World Ocean Atlas 2018
(https://www.ncei.noaa.gov/data/oceans/woa/WOA18/DATA/).

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
