# Peer review of "Elemental Stoichiometry of Particulate Organic Matter across the Atlantic Ocean"

_EGUsphere, 2023_

## Author Comment (AC1)

Reviewer 2:

The authors present new data for the spatial variability in particulate organic matter C:N:P through two latitudinal transects of the Atlantic Ocean with the objective to link patterns in stoichiometry with hydrological conditions and spatial variability in nutrient limitation. This topic that is highly relevant for the readership of many EGU journals. There have been a number of studies published over the last years documenting patterns of stoichiometry both globally and with more regional scope. The work of Fagan et al. contributes a valuable new dataset to this wider documentation of key biogeochemical parameters that are essential to our understanding of marine biogeochemical cycles and how they may change under climate change. Noteworthy is their use of a relatively new metric to interpret areas of N- and P-limitation through the frequency of occurrence of specific marker genes in the highly abundant plankton *Prochlorococcus*. Overall, the manuscript describes well the spatial patterns in C:N:P and nutrient limitation as they relate to hydrological conditions. However, I feel that the authors have missed multiple opportunities to contextualise the motivation of their research and, in particular, their methodology, especially considering their use of a relatively new metric that will not be familiar to many readers. Greater emphasis on the mechanisms and processes explaining their observations would really strengthen the manuscript and its conclusions. My detailed suggestions for improvements are below.

We thank the reviewer for their constructive suggestion on how to enhance the connections of concepts throughout the paper to increase the readability. We also agree that further elaboration upon temperature, nutricline depth and metagenomics is necessary for reader understanding. We recognize that the factors used in our study were not introduced until they were either implemented into the analysis or were given a brief mention in the introduction. We will correct this by introducing these factors within the introduction, specifically how they relate to this study, to provide stronger support for our hypotheses. Along with the introduction, the metagenomics aspect of the study will be elaborated upon within the methods. This will be done by using more concise language and providing greater detail on how they were measured and implemented within this study. We hope that such edits will align with how we and the reviewer view the relationship between these factors.

**General comments:**

1. In my opinion, there are missed opportunities in the Introduction to provide broader context for the study, particularly as pertains to explaining the expected results stated in your hypotheses. The Introduction is generally very brief. There isn't a clear reasoning given for investigating N and P stress patterns through metagenomics vs. other approaches and the introduction would be a great place to line up the need for this relative new approach in filling a knowledge gap beyond what is possible with existing methods (and demonstrating the potential of metagenomics data for addressing challenging areas of research on this topic). The relevance of investigating spatial patterns in N- and P-stress alongside spatial patterns in C:N:P should be explicitly introduced, especially as the introduction currently emphasises physiochemical drivers of C:N:P much

more so than biological drivers. For instance, highlighting that the elemental composition of phytoplankton is responsive to nutrient supply through changes in gene expression and macromolecular composition and that there is evidence for strong changes in stoichiometry under both N and P starvation in representative species of (presumably) all major phytoplankton groups (with citations to appropriate literature). Drivers of C:N:P are generally stated (temperature, nutricline depth) but the processes underpinning their influence on C:N:P (direct and indirect) are not outlined or explored to really any degree in the introduction (see also comments below) (e.g. Lns 76-78 – "Temperature and other environmental factors are also important for C:N:P variability…" with no further exploration of that statement). This leads to a bit of a disconnect between the background of the study and the approach of the data analysis and interpretation.

The reviewer brings up an important point and an oversight on our part. This study focuses on the interactions between temperature, nutricline depth, metagenomics, and stoichiometry. Along with a lack of introducing the use of N vs P stress patterns, we fail to fully describe the known relationship between temperature and nutricline depth with stoichiometry. With future edits to the introductory section, especially an expansion of the third paragraph of the introduction, we can bridge the gap between the introduction (what the reader will be expecting to read in the paper) with the results and discussion.

1. Given the role of *Prochlorococcus* metagenomic data within the study and that the authors have alluded to previous critique of this aspect of the study, I am somewhat surprised that the methodological section on the use of gene content/expression in *Prochlorococcus* as a proxy for nutrient limitation (Section 2.4.4) is in fact very brief. The application of *Prochlorococcus* gene markers as indicators of nutrient limitation is still relatively new and is likely to be less familiar to readers but is an exciting new approach that should be able to enrich existing methods and datasets and hopefully provide new insights into global patterns of stoichiometry – yet nothing about the method is mentioned in the introduction. One issue is that there is no consistent usage of a well-defined and accurate term for this "metagenomics-informed nutrient limitation". The terms 'element-specific nutrient stress', 'gene index', 'severity of nutrient stress', 'nutrient gene index', 'genetic index of nutrient limitation' are all used within a few lines of each other in Methods section 2.4.4 but whether these terms are actually interchangeable is not clear and the description provided of the 'gene index' on Lns 197-200 is still vague and should not, in my opinion, be a 'Briefly,…' type remark. Additional context is required in this section of the methods. For example, the authors could state that there is a close relationship between genome content and local nutrient conditions (with supporting references), that *Prochlorococcus* cells upregulate or actively gain/lose specific genes under P- and N-stress (with gene names and supporting literature), what timescale of response can be expected for the expression or regulation of these genes, and, crucially, that the gene content of cells is therefore an accurate reflection of both the type and severity of physiological nutrient stress experienced by cells in-situ (with supporting literature). This

information, in a clear and easily digestible format, would also be a great addition to the introduction as it is still a relatively unusual approach to the topic of spatial patterns in C:N:P. Either here in the Methods or in the Introduction (e.g. Lns 71-80) or where the study objectives and hypotheses are outlined (Lns 81-88), you could consider adding a clear and explicit statement about what is gained from having nutrient limitation information (through the gene index/proxy) when trying to understand spatial variability in stoichiometry. More specific qualification of the 'significant overlap' between the 'index' and whole community nutrient addition assays (Lns 201-202) would also be a good addition in Section 2.4.4, as it would add more confidence for readers that your approach produces comparable results to more established and familiar (and less 'abstract', i.e. proxy) approaches that readers are likely to be more familiar with.

The reviewer brings up several valid points with this comment. The intention of keeping this methodology brief is because we are using a subset of data from another paper. Further information on the methodology used can be found there. However, we acknowledge that this section was too brief and should provide enough information to the reader that they would not need to read another paper to understand the results of this one. As mentioned in the previous comments the omission of metagenomics in the introduction will be rectified in future edits to provide stronger justification for our approach in addressing our research questions. The reviewer provides several options to better improve the connection of metagenomics to the introductions that will be incorporated with the edits of the third introductory paragraph.

With respect to the terminology issue, especially within the methods, we agree that this is an area of the paper we need to address. To keep the section brief, we moved through this quickly without defining the terms essential to understanding the methodology. We will better define their terms in future edits, along with a more in-depth explanation of the methodology.

1. Building on these points and given the use of a proxy for nutrient stress alongside raw measures of stoichiometry, it is somewhat surprising that there is actually very little discussion about patterns of nutrient stress and the implications of nutrient stress of spatial variability in C:N:P in the Discussion. There is no connection in the Discussion or elsewhere between nutrient stress as a physiological state and the C:N:P content of organic matter (except Ln 73 to say that P use is frugal in P limited regions) in *Prochlorococcus* or any other phytoplankton. Nor are the implications of widespread nutrient limitation for biogeochemistry and ecosystem processes really explored in detail (with the exception of a brief comment on Ln 365-366 that C:N:P variability can 'buffer' the effects of stratification and reduced nutrient supply on primary productivity and carbon sequestration). What are/might be the consequences of the observed latitudinal patterns in stoichiometry for global nitrogen fixation, primary production and carbon

sequestration (Ln 375-376) that are, presumably, a motivating factor in conducting this study? How do changes in the dominant and/or co-limiting nutrient tie into these processes? I feel that the authors have missed opportunities in their discussion to explore these topics in any depth, despite the fact they are directly relevant themes of their research. There is good coverage of the role of N fixation in the North Atlantic and its drivers in Lns 323-334 but beyond that, casual links between observations and process are quite limited in the Discussion. Instead, there is greater focus on restating the main results in the context of hydrology rather than actually explaining the connections between these features (e.g. Ln 342-344 "These zonal shifts in C:N:P can be explained by shallower nutricline depth and stronger N limitation..." – why does shallower nutricline depth and stronger N limitation explain different C:N:P? i.e., why does greater/less nutrient supply change C:N:P and what is that dependent on? How does N limitation actually influence C:N:P and why, mechanistically? Is it the same for all phytoplankton groups?).

While we had discussed the components of the research in the discussion, we had not elaborated on how these findings connected between the stoichiometry and environmental factors. The first paragraph of the discussion focused on variability of stoichiometry to other research cruises, providing further support to systematic biome shifts. The following paragraphs focus on the environmental factors influencing stoichiometry but fail to explain the connections between them. Future edits on these paragraphs will further elaborate on the causal links between these features. For the influence of nutricline depth and temperature, we will elaborate on the influence of nutrient input to the surface and the differences on the eastern and western boundary currents. Western currents typically being warmer and saltier than the eastern, causing an increase in stratification and limiting the availability of nutrient to the surface. Also, with warmer conditions phytoplankton reduces the number of ribosomes they build as they are more efficient compared to colder temperatures. A decrease in ribosome production will lower the demand of phosphorus by the phytoplankton, increasing the N:P and C:P ratios. Using the nutrient gene index, we can explain how some of the variability observed with C:N:P could be the result of phytoplankton (*Prochlorococcus*) expressing these specific genes that allow for them to uptake a greater amount of a nutrient than if they do not express the gene. This can be observed from the hemisphere difference when using either nutrient gene index; high N being greater in the South lead to a greater C:N than in the North and vice versa with high P. As some of these features are universal to major ocean basins, it is possible to then apply these observations to other oceans, especially those under sampled.

1. There are no remarks in the discussion concerning the role of differences in C:N:P between different phytoplankton groups and their uptake strategies and the spatial patterns in C:N:P observed, beyond saying that the abundance of nitrogen fixers in the North Atlantic can drive P-stress. This seems like an obvious omission, especially considering that the authors have at least partially quantified the dominance of different phytoplankton groups in each region, shown in Figure S4 (and presumably have this data on a site-by-site basis

too). For instance, in the regions/sites, where *Prochlorococcus* is particularly dominant (numerically and/or in biomass), what is the C:N:P of *Prochlorococcus* (under these environmental conditions) relative to the stoichiometry of the other taxa present? Is there therefore a link between the taxonomic composition of the community (and the POM) and the stoichiometry of the different groups present? If it is not possible to analyse this relationship, even roughly, then something to this effect (that is cannot be determined and why) can be added at relevant sections of the manuscript.

The reviewer makes a fair point about focusing on *Prochlorococcus* rather than the rest of the phytoplankton groups. We unfortunately are unable to assess how the C:N:P would vary between each group as the POM samples are an accumulation of all the groups. We see the value in acknowledging this lack of information as being important to a reader and a potential step for future projects.

**Specific comments:**

Ln 65: add what Redfield proportions of C:N would be.

This will be corrected in future edits.

Ln 62-63: In the Indian Ocean and Pacific Ocean cruise data, what patterns in C:N:P, POM and environmental gradients were observed? How were these explained? Would you expect patterns to be similar and with similar causes in the Atlantic? If not, why not? Addressing these points (and those below) would provide more complete context for the reader and better support your hypotheses.

The patterns for the two cruises are described in the discussion section of the paper, when comparing the findings. We agree that this information should be presented up front in the introduction, so that the readers have the same understanding going into this paper as we did. Future edits will include this information within the second introductory paragraph. This will pair with the hypothesis we present with the first research question of this paper, which will also be added.

Ln 71-72: the wording of this sentence is quite vague. This can be read to mean that everything in the northern hemisphere experiences phosphorous limited and everything in the southern hemisphere is nitrogen limited, which I am sure is not actually accurate. A more specific geographic description would be more accurate and informative here. You could also mention here what processes have been used to explain this pattern in nutrient limitation.

We agree with the reviewer and in future edits will provide more specific descriptions of the regions in question.

Ln76-88: Whilst you describe the patterns of nutrient limitation in the Atlantic Ocean and relationships between environmental conditions and C:N:P generally, you say very little in the introduction to explain why these relationships may exist in these regions. The only reference to biology is Ln 52-53 and all that is stated is "…observed variability in marine plankton and ecosystem elemental composition." I think that your introduction would be supported by including some additional details of the specific biological and/or physiochemical processes that have been used to explain these patterns of nutrient limitation in the literature that you already cite and the reason why "temperature and other environmental factors are also important for C:N:P variability" (Lns 76-77). For example, you could introduce that P-limitation in the North Atlantic has previously been linked to the higher productivity of nitrogen fixers (i.e. increasing N:P) that thrive under the higher iron availability supplied from the Sahara region. By not exploring any reasons behind any of the drivers of C:N:P, there is little justification for why you hypothesise that nutrient supply and temperature are primarily responsible for latitudinal variability in C:N:P, beyond the inference that there is a relationship reported in the literature. At least some of the variability in C:N:P can be attributed to changes in phytoplankton community composition and the changing abundance of groups that have very different nutrient acquisition and utilisation strategies, but this is effectively not even mentioned in the introduction (with the exception of "frugal P use" Ln 73, but it is not clear to the reader if this is a universal strategy).

We agree that this section is too surface level with the information we are presenting to the reader. Along with this we feel that the section could be better used in presenting the factors that we investigate later in the paper, as they are essential components to the presented hypotheses. Significant future edits will be made to this section to justify our use in temperature, nutricline depth, and nutrient gene stress. Community composition does have some influence on the variability of C:N:P, however, this work primarily focuses on the oligotrophic ocean that is dominated by *Prochlorococcus* (both in count and biomass).

Ln 83: You state that you have address three questions but only two questions are given.

This will be corrected in future edits.

Ln 93-96: Perhaps mention the map of the cruise transects shown in Figure 1 in this section of the Methods?

This will be corrected in future edits.

Ln 105-106: What 'large particles' would typically have been removed using this mesh size? A significant amount of phytoplankton are >30μm including most diatoms, diazotrophs, dinoflagellates etc. It is fair to say that you skew your C:N:P towards nano- and picoplankton C:N:P? If this is a deliberate choice, then this should be justified here and explicitly stated what the constituents of the measured C:N:P is largely representative of. If this is a standard collection procedure already used, relevant literature should also be cited here (especially as in the introduction Ln 60 you specifically highlighted that an aim of the Bio-GO-SHIP cruises was to

utilise consistent methodologies). How might the choice of a 30μm prefilter and the exclusion of larger cell sizes (and perhaps a large proportion of certain taxonomic groups) impact your results? This could be mentioned here in the method or as a paragraph in the discussion.

A majority of the of the transect that samples are collected have phytoplankton that are smaller than 30 μm. Comparisons have shown that cells larger than 30 um rarely constitute more than 10% of biomass. In oligotrophic regions, POM concentrations with and without this filter are often indistinguishable. Lee et al. 2021. Linking a Latitudinal Gradient in Ocean Hydrography and Elemental Stoichiometry in the Eastern Pacific Ocean showed this in detail. However, for consistency we have used this size filter on all of the previous cruises and if it turns out there is a significant percentage to the amount of phytoplankton we are missing, then it can be addressed as a whole. In future edits we will cite past papers that have used this method.

Ln 120: bias of what specifically?

This will mitigate bias of phytoplankton growth throughout the day.

Ln 156-157: is there data support for this nutrient supply proxy/nutricline depth that could be referenced here?

We will add citations in future edits to support the use of nutricline depth as a proxy to nutrient supply.

Ln 181-190: Are the cell size samples also from water samples that have been pre-filtered through a 30 μm mesh?

These samples have not been pre-filtered through the 30μm mesh, and we will clarify that in future edits.

Ln 194-204: This section of the methods is very important and uses a relatively new and novel methodology that may not be at all familiar to many readers. Because of this, it is vitally important that this section is as clear and explanatory as possible. However, I find the description of the gene/nutrient index to be written quite unclearly relative to other sections of the methods (although I appreciate that this may be because I am not as familiar with this approach as I am with other parts of the methodology). For instance, what is meant specifically by "element-specific nutrient stress was used…" in Ln 194? 'Element-specific nutrient stress' is not really defined in this context and it is not at all clear what was 'used' from the global genome content of Ustick et al. (2021). Does this mean that the data analysis uses the exact same dataset used by Ustick et al. (2021)? Or just a specific subset of this data? Or were new metagenomic samples run from AMT-28 and C13.5 cruises for the purposes of this study? Again, when you say 'the described metagenomic samples' do you mean here the samples previously used in Ustick et al. (2021)? You also mention that you used just the information representing the 'most severe form of the nutrient gene index' but you have not provided any information on how magnitude of nutrient stress is expressed or identifiable in the gene index and therefore which specific genes or information you have used. It is essential that these

points are further clarified. Is it the case that there is a progressive sequence of the expression or upregulation of genes for different P acquisition strategies as P-availability decreases (i.e. becomes more stressful), and therefore depending on which specific genes or combination of genes are expressed you can infer whether P-/N-availability is causing moderate vs. severe nutrient stress? From briefly reading Ustick et al. (2021) and Table 1 of their study, my interpretation would be that by focussing on severe stress you would be looking at the content of the genes *phoA* and *phoX* related to alkaline phosphatase and not genes related to P-starvation etc. that are more indicative of 'medium' P-stress. If this is the case, then something to this effect within this section is necessary. Quantitatively, it seems from the caption of Figures 4 and the axes of Figure S6 and S9 that the gene index has a numerical value ranging across positive and negative values. At a minimum, there should be a description of the 'calculation' of these values (my understanding from the caption of Figure 4 is that is it a type of frequency of occurrence or abundance measure) and for easy interpretation, what a negative vs positive value indicates (if anything) and how large the difference between, say a value of 1 vs. 2 of the gene index should be interpreted, i.e. does 2 indicate double the nutrient limitation of 1? Your methodology needs to be completely understandable without having to refer to other papers.

We understand and agree with the points that the reviewer brings up here. An important addition to add to this section will be a description of key terms that we then start to use throughout the paper. This will prevent any assumptions about the terms we used. Future edits will improve on the consistency of terms as we had on occasion misused a term or used a new term without describing it to the reader. This study uses a subset of data from Ustick et al., (2020), specifically the samples AMT28 and C13.5. Significant edits will be made in the future to this section of the methods.

Ln 257: N* is introduced here for the first time but you start describing the pattern of N* before defining what it is/what is represents, how it is interpreted and therefore why positive or negative N* values are indicative of P-limitation or N-limitation, respectively. There is also no reference supporting the origin of this parameter, although I do recognise that it is relatively widely used in the literature, it may not be familiar to all readers, and it should still be referenced and defined properly. This definition could also be added to the caption of Figure S3. Although it is mentioned here in the Results, there is no further exploration in the Discussion of why there is only a weak agreement between N* and N:P ratios in your data and what the cause of this might be. As N* has been mentioned here as being another indicator of primary limiting nutrient, it would also make sense to say something in the introduction and/or methods that the calculation of N* (including defining what it is) is one existing approach for inferring areas of N and P limitation and why you have not used this as your principle nutrient limitation indictor in this study and have instead opted to use the *Prochlorococcus* gene-based nutrient limitation proxy. Would you expect both methods to give comparable results? If not, why not?

While N* might be a term widely used in literature, we agree that the term should be properly defined/ described. In future edits we will define N* before its incorporation within the text of

the paper and Figure S3. Several lines within this paragraph will be added to describe the use of N* as a proxy and how the results should compare to gene-based proxy. However, further discussion of the end results will be covered in the discussion section.

Ln 265-266: A sentence or two saying what you allude to here in this sentence, i.e., 'phytoplankton community composition, temperature, nutricline depth and nutrient stress are all possible drivers of stoichiometry', is missing from the introduction and should be included with further context, e.g. why is temperature a driver of stoichiometry? Effects on cell physiology and/or changes in macromolecular content? A link between temperature and dominant phytoplankton group relative to the ecological preferences and/or productivity of different groups? A link between temperature and stratification and therefore nutricline depth? All of the above interacting with additional factors too?

We recognize that we did not elaborate on the relationship of these factors to stoichiometry earlier in the discussion, but we did mention there being a relationship with temperature and nutricline. We agree that it is important the reader is presented with these concepts earlier on to understand the direction of the paper. In future edits we will add phytoplankton community composition and nutrient stress factors and their relationship to C:N:P in the introductory section of the paper. Based on the current structure of the introduction, this information will be added in the third paragraph.

Ln 265-273: You only mention *Prochlorococcus* data here. Especially in the areas where *Prochlorococcus* was not dominant numerically or in terms of biomass, which group(s) were the other major contributors? It is a shame that the figures of assemblage composition are only supplementary figures rather than in the main manuscript – as a potential driver of the observed patterns in stoichiometry alongside nutricline depth, temperature and nutrient stress, should this Figure not be given equal space in the main manuscript figures? Is there a reason why you have made broader groupings of the phytoplankton types in Figure S4 compared to your description of the categorisation of photoautotrophs described in the Methods Ln 185-186? The category 'Other eukaryotes' is a large or majority contributor to total biomass in all regions but there is no further information of which taxa are present in those areas (and how differences in those taxa may contribute to your results). If the data presented in Figure S4 represent the >30 µm community composition only (see previous comment about the Methods) then this should be added to an appropriate part of the Results section and to the caption of Figure S4.

The focus on *Prochlorococcus* in this study is because we only have *Prochlorococcus* metagenomics. Figure S4 is a supplementary figure because we are primarily using it as part of the argument as to why *Prochlorococcus* is a viable representative of the different regions. Since we are working to explain the variability of POM and stoichiometry across the Atlantic and the relationship of environmental factors and stoichiometry, we felt that Figure S4 should be a supplementary figure. We agree with the importance of this figure, but do not feel that this figure alone helps to answer our research question. The main reason for this categorization is primarily for simplification. There was no fourth taxa that was consistently a large portion of

each graph, so we felt that this would be enough for the readers to get an understanding of the count and population breakdown. Yes, the other eukaryote section is a large contributor to the population biomass, and dominant for ~10% of the transect, but we are primarily focused on *Prochlorococcus*. This data was collected without the 30µm mesh, and we will clarify this in future edits of the methods section. Additionally, the category of other eukaryotes will be given in the figure caption for readers wanting to know more.

Ln 285-286: I think it is important to at least list some suggested additional factors that account for the total deviance explained (with supporting literature), as in Figures 4, S7 and S8, over 30% of C:P and N:P and almost 80% of C:N is explain by 'other' factors, so it is definitely not an insignificant amount.

The other factors we found did not have a direct influence on the uptake of N or P. Or the measurements of the other factors would not be accurate enough to give results we are confident in. Additionally, with this being in the results section of the paper, we aim to minimize the amount the discussion added. We do see value in adding some discussion of these factors in the discussion section though. In future edits we will leave the section referenced above alone or add a line of further discussion in the discussion section and add the points of addition factors to the discussion section with reference that support their potential influence in C:N:P.

Ln 294: typo

This has been corrected.

Ln 302: Is this a reasonable assumption? Or purely a necessary one. If so, maybe you can add 'in the absence of seasonal data for P- and N-stress from our nutrient stress proxy, we assume that the biogeography of N- and P- stress remains stable year-round' or something to that effect.

The reviewer has made a fair speculation and is correct to ask. Since we lack any seasonal data for the nutrient stress we need to assume a stable biogeography of this factor. In future edits we will work to clarify this point. Like breaking this line into two sentences, one with the seasonal factors and the other with being in the line of; In the absence of seasonal data for the nutrient gene index, we assume that the biogeography remains stable.

Ln 323-334: Is the link between iron inputs, nitrogen fixation and spatial C:N:P patterns for the Atlantic Ocean supported by observations in the Pacific and Indian Oceans or elsewhere? Even supporting literature for regional studies where there might be similarly high iron inputs locally, leading to high abundance of nitrogen fixers and correspondingly skewed N:P ratios, etc. You state in Lns 318-320 that similar gradients to your results have been seen in other ocean basins, but you do not refer again to whether the processes involved in the Atlantic are similar to those in other ocean basins or not.

There is a study by Garcia et al., (2020) that investigates the link between iron inputs, N-fixation, and C:N:P in all three basins. We agree that this section could be improved by further references that support our finds and how they might relate to other ocean basins.

Line 318-320 are meant to provide contexts that the results we found in this study do not differ from our expectation, based on the findings of the past cruises. However, we feel the reviewer brings up a good point that this section could be better supported by discussing the regional similarities. One potential version would be a brief mention of the temperature and nutricline depth patterns across the basin. It is important to note that this paper is focused on the Atlantic and not meant to act as a synthesis paper comparing the regional similarities and differences between the basins.

Reference: Garcia, C.A., Hagstrom, G.I., Larkin, A.A., Ustick, L.J., Levin, S.A., Lomas, M.W., Martiny, A.C., 2020. Linking regional shifts in microbial genome adaptation with surface ocean biogeochemistry. Philosophical Transactions of the Royal Society B: Biological Sciences 375, 20190254. https://doi.org/10.1098/rstb.2019.0254

Ln 336-337: the reference to Ustick et al. (2021) here is the (presumably) identical method and perhaps identical dataset (not clear from methods, see previous comments) to your results. Can you add additional literature citations for the prevalence of P-limitation in the North Atlantic based on other types of evidence? Or at least state that this is based on the same data/proxy. Same for Lns 343-344.

The dataset used in this paper is the same as Ustick et al., (2021). Future edits will be clearer about the data used in this paper. We will address this in both the discussion section and the methods.

Ln 338: Edit to 'Using nutricline depth as a proxy for magnitude of nutrient availability....' or something similar in order to be more precise.

Describing it as a proxy in this line redundant. We described the use of nutricline depth as a proxy for nutrient supply to the surface in the methods section. In this instance, we are referring to this definition that has been used previous studies (two examples Garcia et al., 2018 and Moreno et al., 2022).

Garcia, C.A., Baer, S.E., Garcia, N.S., Rauschenberg, S., Twining, B.S., Lomas, M.W., Martiny, A.C., 2018. Nutrient supply controls particulate elemental concentrations and ratios in the low latitude eastern Indian Ocean. Nat Commun 9, 4868. https://doi.org/10.1038/s41467-018-06892-w

Moreno, A.R., Larkin, A.A., Lee, J.A., Gerace, S.D., Tarran, G.A., Martiny, A.C., 2022. Regulation of the Respiration Quotient Across Ocean Basins. AGU Advances 3, e2022AV000679. https://doi.org/10.1029/2022AV000679

Ln 347-354: Is *Prochlorococcus* known to show the same stress response as other phytoplankton groups? I.e. if Prochlorococcus is P-stressed then will the concentrations of P also be low enough to be considered stressful for all other taxa in the community? An explicit statement to this effect would be probably more relevant than saying that *Prochlorococcus* is a good 'starting point' purely because it is very abundant in the subtropical gyres.

The reviewer brings up an important question about the use of *Prochlorococcus* that we had provided minimal support for in our usage. Prochlorococcus biomarkers correlate very well with both nutrient addition experiments, model simulations and other information about the regional shifts in nutrient stress type. Furthermore, it is the smallest phytoplankton so if Prochlorococcus is limited by a nutrient, it is likely that other lineages are as well. However, we agree on the sentiment and did not intend to say that we know for certain that all community members experience the same stress as Prochlorococcus. In future edits, we will try to make this point clearer, within the methods, results, and discussion sections.

Ln 371-372: Can you add to this any supporting evidence from other areas of iron scarcity or enrichment for the role of iron supply in regulating C:N:P? The paper of Ustick et al. (2021) also mentions gene markers for iron stress – is there any additional information from the presence of these markers in *Prochlorococcus* that could further expand on this section of the discussion, albeit briefly?

There are other studies that investigate the role of iron scarcity with *Prochlorococcus,* within the Atlantic Ocean and other ocean basins. While there is discussion within the paper about the influence on iron, we have held back from including the Ustick et al., (2021) iron data as, iron has a more indirect influence on the uptake of nitrogen and phosphorus compared to the other data presented in the paper.

**Further comments:**

Note that the front piece of your supplementary file is not formatted to the correct journal!

This will be corrected in future edits.

Figure S6, S9: what exactly is being plotted here on the y-axis? There is no explanation here or in the methods what "(average …) gene stress" refers to or how it is quantified to give a numerical value such as you have plotted (there is a better caption description in Figures S7 and S8). It is therefore, for instance, impossible to interpret whether a difference in value between 0 and 2 "high P stress" is a large magnitude of difference or not between regions and what that actually means. Perhaps a more descriptive axis label than "High N stress" and "high P stress" could be used.

The figure is meant to describe the change in the high nutrient gene index across the Atlantic Ocean. With respect to the usage of high, medium, and low stress types, we had oversimplified in this paper how it was selected. Using the data from Ustick et al., (2021), we focused on the high stress as it indicated adaptations that would be advantageous for *Prochlorococcus* in a low

nutrient (high stress) condition. High and medium stress genes in Ustick et al., (2021) had a near identical pattern and we felt confident showing just the high stress would suffice. The values are unitless as they are meant to represent the variation of nutrient stress across the ocean. In future edits we will work to make this information clearer, as well as address the missing information in Figure S6 and S9.

**References**

Ustick LJ, Larkin AA, Garcia CA, Garcia NS, Brock ML, Lee JA, et al. (2021). Metagenomic analysis reveals global-scale patterns of ocean nutrient limitation. Science **372**: 287–291.

**Citation**: https://doi.org/10.5194/egusphere-2023-2453-RC2

---

## Author Comment (AC2)

The authors provide data on elemental ratios of organic matter from the Atlantic Ocean. The data is highly valuable, and thus EGUsphere seems suitable for this manuscript. The measurements from two different transects provide insights into both latitudinal and longitudinal variation. The authors explore the relationship between the elemental ratios and environmental factors. One major concern is the description of nutrient stress and nutrient limitation. The entire manuscript seems to rely on genomics (of Prochlorococcus), which does not match the nutrient limitation data based on the established method. I suggest that the authors clarify this discrepancy in the main text so that the readers are aware of this limitation. Such clarification is important because the nutrient limitation indicated by the Prochlorococcus only genomics analysis tends to be skewed toward P limitation, creating a misleading impression of nutrient limitation.

The primary concern brought up by this reviewer relates to how we incorporate genomic information about nutrient stress as predictors for differences in C:N:P. The reviewer argues that bioassays show that N is the primary limiting nutrient in many of the places we study whereas we observe high frequencies of P stress genes. The issue of which element limits growth and productivity in different regions is a very tough question that we do not intend on solving in this study.

We recognize that we were inconsistent in the presentation of our biomarker data and used the term limitation. This will be corrected. Instead, the combination of metagenomic biomarkers of nutrient stress and ecosystem C:N:P suggest that when ecosystems are 'stressed' by a particular nutrient, C:N:P changes. This is consistent with culture and community experiments showing that C:N:P is very sensitive to changes in nutrient availability. Independent of which element ultimately control growth, stressful conditions can affect the resulting C:N:P. We aim to more carefully delineate this argument and ensure that we are only considering stress conditions and not outright nutrient limitation. This will be done by introducing this difference between stress and limitation in the introduction, more careful use of terms in the results, and then revisit the issue in the discussion. We hope that such edits will align with how we and the reviewer view these important biological and biogeochemical controls.

L72:

Regarding P limitation, the prediction from an established method shows that it is a secondary limitation (Moore et al., 2013). It might be good to clarify that in these regions, N is the main limiting factor. The paper shows that P does not come mainly as a main limitation. A recent study shows P limitation-related genes across the ocean, but having related genes might be different than the actual limitation on organismal growth.

You are right that there is an established methodology that supports nitrogen as the primary limitation and phosphorus as the secondary. This is supported by more recent papers – e.g., Browning and Moore, 2023. It was not our intention to have the paper present phosphorus as the limiting nutrient for the region, rather that cells are stressed by phosphorus and thus altering C:P. Culture experiments show that elemental ratios are very sensitive to nutrient

stress – even in the absence of overall biomass accumulation being limited. Thus, we plan to summarize and discuss our hypothesis for how nutrient stress impact ecosystem C:N:P.

L266:

Genomics may not necessarily represent the nutrient limitation: having genes is different than the actual growth limitation. Whether the genes are used to compensate for nutrient limitation is not clear with genomics analysis. Likely because of that, the genomics and the actual limitation seem very different (compare Ustick et al., 2021 with Moore et al., 2013). I suggest that the authors explicitly state this discrepancy in the manuscript to reduce misleading impressions.

 We agree and will carefully describe that biomarkers indicate that cells are 'stressed' by a particular element.

L269:

>93%: I suggest the authors clarify this is based on the cell count. I see that Fig. S4 has it, but clarifying this in the main text would help readers understand the number.

 Rereading this section, I can understand the confusion in the percentage. There are a few places in this paragraph with a similar structure that we will clarify.

E.g., From this *Prochlorococcus* was determined to make up 93% of the community from cellular counts in the subtropical gyres and equator and contributing to over 50% of the total biomass in those same regions.

L268-273

Fig. S4 has Synechococcus in it, which I found valuable information. I hope that the authors describe it in the main text.

 While the data we have available does include Synechococcus counts, we do not have corresponding genomic data. We find that the addition of Synechococcus might cause some confusion, as to why they are brought up when the primary focus is *Prochlorococcus*. This is still an interesting point, however. A paper by Garcia et al., 2020 uses genomics from several cruises to compare *Prochlorococcus* and *Synechococcus*. They found that they follow the same trends across the transect as each other. This will be clarified in the revised version.

Reference: Garcia, C.A., Hagstrom, G.I., Larkin, A.A., Ustick, L.J., Levin, S.A., Lomas, M.W., Martiny, A.C., 2020. Linking regional shifts in microbial genome adaptation with surface ocean biogeochemistry. Philosophical Transactions of the Royal Society B: Biological Sciences 375, 20190254. https://doi.org/10.1098/rstb.2019.0254

L280-282

As mentioned above, there is a discrepancy in nutrient limitation between the metagenomic estimate and the established methods. I suggest this point is clarified somewhere in the text. For example, the established methods show N as a key limiting factor (and P as secondary), and the result in this present paper may not represent the actual growth limitation.

We agree and will clarify that cells are 'stressed' by a particular element. See also the earlier comment for details on this point.

Fig. 4

Because the nutrient limitation is based on the metagenomics analysis, this result could be misleading. I suggest that the authors make clear the difference between the actual growth limitation and the prediction of nutrient limitation based on the metagenomics analysis. For example, Moore et al. 2013 compiled the results of nutrient incubation analysis, resulting in N as a primary limitation in the North Atlantic. Given that, this figure seems to overemphasize P limitation because it is based on metagenomics, and I suggest that the authors make clear the caveats (especially the inconsistency with the outcome of the established methods) of the metagenomics analysis somewhere in the text.

We plan to carefully discuss the role of nutrient stress vs. growth limitation and what it means for the regulation of ecosystem C:N:P. We believe this will reconcile the differences between bioassays and metagenomics and still provide important insights into differences in ocean C:N:P.

L297

Here, genes may not tell nutrient stress. For example, in culture studies, organisms with the same gene may experience various nutrient stresses regardless of genes. Genes could be a proxy, but as mentioned above, there seems to be a clear discrepancy between the established methods and estimates from genes. I suggest using terms such as "stress proxy," "stress indicator," or, more explicitly, "stress-related genes."

Now I noticed that Figure 4 uses the term "nutrient gene index." I think it is a good expression, and the term is well defined. I suggest including such a definition in the main text as well and using the term throughout the paper instead of simply saying "nutrient stress" or "nutrient limitation" because, apparently, these are different things.

 We agree that there is an inconsistency with the terms we use, and that they were used interchangeably. Using the term "nutrient gene index" we can prevent confusion as to which nutrient stressor we are referring to. We also agree that in future edits we will take care to standardize the terms we use and to give them a well-defined definition that remains

consistent through the paper. Along with nutrient gene index we will also make sure that the other terms referred to will be corrected and/or defined.

L335-337

Please see the earlier comments. These may not be actual P limitations, so I suggest clarifying this a bit more. e.g., shift from N stress genes toward P stress genes.

We agree with this statement and will correct it in future edits.

L343

Similarly, stronger P limitation may not be accurate. I suggest rephrasing (see above).

We agree that the use of limitation was too broad and should be narrowed down to a more accurate statement. i.e nutrient stress shift based on the nutrient gene index.

L347 "N and P-limitation" Please see the above comments.

This will be corrected in future edits.

References

Moore CM, Mills MM, Arrigo KR, Berman-Frank I, Bopp L, Boyd PW, et al. (2013). Processes and patterns of oceanic nutrient limitation. Nature Geoscience **6**: 701–710.

Ustick LJ, Larkin AA, Garcia CA, Garcia NS, Brock ML, Lee JA, et al. (2021). Metagenomic analysis reveals global-scale patterns of ocean nutrient limitation. Science **372**: 287–291.

**Citation**: https://doi.org/10.5194/egusphere-2023-2453-RC1